# The Impact of Pruritus on the Quality of Life and Sleep Disturbances in Patients Suffering from Different Clinical Variants of Psoriasis

**DOI:** 10.3390/jcm11195553

**Published:** 2022-09-22

**Authors:** Kamila Jaworecka, Marian Rzepko, Luiza Marek-Józefowicz, Funda Tamer, Aleksandra A. Stefaniak, Magdalena Szczegielniak, Joanna Chojnacka-Purpurowicz, Ayla Gulekon, Jacek C. Szepietowski, Joanna Narbutt, Agnieszka Owczarczyk-Saczonek, Adam Reich

**Affiliations:** 1Department of Dermatology, Institute of Medical Sciences, Medical College of Rzeszow University, 35-310 Rzeszow, Poland; 2Institute of Physical Culture Sciences, Medical College of Rzeszow University, 35-310 Rzeszow, Poland; 3Department of Dermatology and Venerology, Faculty of Medicine, Nicolaus Copernicus University in Toruń, Ludwik Rydygier, Collegium Medicum in Bydgoszcz, 85-094 Bydgoszcz, Poland; 4Department of Dermatology, Gazi University School of Medicine, 06570 Ankara, Turkey; 5Department of Dermatology, Venerology and Allergology, Wroclaw Medical University, 50-367 Wroclaw, Poland; 6Department of Dermatology, Pediatric Dermatology and Oncology, Medical University of Łódź, 92-215 Lodz, Poland; 7Department and Clinic of Dermatology, Sexually Transmitted Diseases and Clinical Immunology, Faculty of Medicine, Collegium Medicum, University of Warmia and Mazury in Olsztyn, 10-720 Olsztyn, Poland

**Keywords:** psoriasis, palmoplantar pustulosis, pruritus, itch, itching, quality of life, quality of sleep, sleep disturbances, sleep disorders

## Abstract

Background: Quality of life (QoL) and sleep, which are essential for well-being in the mental, physical, and socioeconomic domains, are impaired in psoriatic patients. However, the exact role of the clinical subtype of psoriasis in this aspect remains poorly studied. Objectives: The aim of this study was to investigate differences in QoL impairment and sleeping problems in patients suffering from various clinical subtypes of psoriasis and to evaluate the effects of pruritus on QoL. Methods: This cross-sectional, multicenter study included 295 eligible subjects with diagnosed psoriasis. Each patient was examined with the use of the same questionnaire. Measures included predominant subtype of psoriasis, disease severity, pruritus scores, patients’ health-related QoL and the incidence of sleep disturbance. Results: The QoL of most patients was decreased irrespectively of clinical psoriasis subtype, however, the most impaired QoL was in patients with erythrodermic psoriasis. The majority of patients reported sleep disturbances caused by pruritus, albeit there was no relevant differences between analyzed subgroups in this aspect of patients’ well-being. Pruritus was an important factor determining QoL and sleeping problems in the studied population. Conclusions: Identifying the most disturbing area of life and recognizing the most bothersome subjective symptoms of psoriasis are pivotal to focusing on the most relevant treatment goal and achieving therapeutic success.

## 1. Introduction

Psoriasis is a highly prevalent, chronic, inflammatory disease, primarily affecting the skin. The most common skin lesions are erythematous papules and plaques covered with silvery scales, frequently localized on the exposed skin areas as knees, elbows or scalp. Based on clinical presentation, several variants of psoriasis can be distinguished, such as plaque-type psoriasis, small-plaque psoriasis, guttate psoriasis, inverse or flexural psoriasis, psoriasis of the scalp, hands and feet psoriasis, erythrodermic psoriasis, palmoplantar pustular psoriasis (PPPP) and generalized pustular psoriasis (GPP).

Quality of life (QoL), defined as a complex and multidimensional concept and an index of subjective well-being in the mental, physical, and socioeconomic domains as perceived by individuals, is usually decreased in dermatological patients [1]. Health-related QoL is a term that refers to health status and severity of disease. Many factors such as involvement of visible parts of the body, scaling, or subjective symptoms such as pruritus or pain may have a negative impact on patients’ QoL. Emotional stress caused by attempts to hide the skin lesions and lowered self-esteem negatively affect interpersonal relationships. The situation may be of particular relevance if skin lesions are located within the anogenital area, as this may affect sexual activity and relationship with the partner. Social awareness of this disease is still low, therefore, psoriatic patients are often stigmatized and excluded by the social and work environment. All these above-mentioned aspects contribute to sadness, depression, anxiety, and suicidal ideation [2]. Moreover, stress and negative emotions are one of the best known factors that exacerbate psoriasis; thus, patients need to be able to manage these symptoms in order to prevent further disease worsening [3].

Sleep is undoubtedly an important aspect of general health. An insufficient amount of high-quality sleep may cause fatigue, irritability, emotional lability and, as a consequence, impaired functioning in the society [4]. Many studies reported sleep disturbances in dermatological diseases including psoriasis [5,6,7], but data on sleeping problems in different clinical subtypes of psoriasis are limited.

Here, we aimed to investigate and characterize the differences of QoL and sleeping problems in different clinical variants of psoriasis with a special emphasis put on the role of pruritus, as it is the most frequent and the most bothersome subjective symptom of psoriasis [8,9,10].

## 2. Materials and Methods

### 2.1. Study Design

This cross-sectional study was conducted from June 2020 to November 2021 in six dermatological departments localized in Poland (Rzeszów—*n* = 102, Bydgoszcz—*n* = 52, Wrocław—*n* = 43, Łódź—*n* = 39, Olsztyn—*n* = 20) and in Turkey (Ankara—*n* = 82). The study was approved by the Ethics Committee of the Subcarpatian Physician Chamber in Rzeszow for Poland and by Gazi University Ethics Committee for Turkey. Patients were in details informed about the study aims and signed an informed consent form before initiation of any procedure.

### 2.2. Studied Population

Patients were recruited consecutively from the patients visiting our departments. The inclusion criteria were as follows: each patient had to have an established diagnosis of psoriasis that could be classified to one of the predefined psoriasis subtypes (large plaque psoriasis, small plaque psoriasis, guttate psoriasis, palmoplantar psoriasis, psoriasis of the scalp, inverse psoriasis, erythrodermic psoriasis, PPPP and GPP) and be able to complete the questionnaires. The exclusion criteria included age <16 years, illiteracy, presence of any dermatological or systemic disorder that might cause pruritus, treatment of psoriasis within a period of 2 (for topical agents) or 4 weeks (for systemic therapy or phototherapy) before study inclusion, pregnancy, lactation, and use of drugs with antipruritic potential.

Out of 338 psoriatic patients enrolled primarily in the study, 295 were included for final analysis. The remaining 43 subjects were excluded because of current or recent (within the prior 4 weeks for systemic agents or phototherapy, or within 2 weeks for topical medications) anti-psoriatic treatment or comorbidities of either dermatological or systemic disorders that might cause pruritus.

Patients were divided into nine groups according to the dominant clinical manifestation of psoriasis: large-plaque psoriasis (*n* = 45), small-plaque psoriasis (*n* = 32), guttate psoriasis (*n* = 31), palmoplantar psoriasis (*n* = 33), psoriasis of the scalp (*n* = 32), inverse psoriasis (*n* = 23), erythrodermic psoriasis (*n* = 33), PPPP (*n* = 43) and GPP (*n* = 23) [11]. The majority of the investigated groups were comparable according to gender, age, duration of psoriasis, BMI and coexistence of psoriatic arthritis. However, some statistically significant discrepancies were observed, which can be explained by particular subtype features (e.g., patients with GPP were older, and patients with plaque-type psoriasis were more obese) (Table 1).

### 2.3. Psychometric and Clinical Assessments

QoL was assessed with the Dermatology Life Quality Index (DLQI), which contains ten questions covering different topics related to skin disease such as symptoms, change in habits, hobbies, clothing style, leisure and social activities [12,13,14]. Because of its simplicity and reliable grading, DLQI is widely used in numerous clinical and non-clinical studies. The minimum and the maximum scores vary between 0 and 30, and the higher the score the higher the QoL impairment. Sleeping problems were evaluated using 3 simple questions about problems in falling asleep, frequent awakenings and using sleeping medication.

The 11-point Numerical Rating Scale ranging from 0 (no pruritus) to 10 (worst imaginable pruritus) was used to assess the maximal (NRSmax) and average (NRSaverage) intensity of pruritus during the last three days [15]. The other method of pruritus intensity measurement was the 10-item Pruritus Severity Scale (10-PSS) [16]. In this scale, questions about the severity of pruritus (two questions), duration of itch episodes (one question), localization of pruritus (one question), influence on patient’s concentration and psyche (four questions), and scratching behaviors in response to itching (two questions) were included. 10-PSS scoring ranges from 3 to 20 with a higher score indicating more intense pruritus.

Disease severity was measured with the Body Surface Area (BSA), and the Static Physician Global Assessment (sPGA) in all patients as well as with the Psoriasis Area and Severity Index (PASI) in patients with large-plaque psoriasis, small-plaque psoriasis, guttate psoriasis, palmoplantar psoriasis, psoriasis of the scalp, inverse psoriasis and erythrodermic psoriasis [17,18]. Patients with PPPP were assessed with the Palmoplantar Pustulosis Severity Index (PPSI), while those suffering from GPP according to the Generalized Pustular Psoriasis Severity Index (GPPSI) [19,20].

### 2.4. Statistical Analyses

The Statistica 13.0 (Statsoft, Krakow, Poland) software was employed for statistical analysis. Means, standard deviations (SD), maximum, minimum, median values, and frequencies were calculated. The differences between the comparted groups of patients were verified with the analysis of variance (ANOVA) and with a regression analysis. Correlations between analyzed parameters were verified by using the Spearman’s rank correlation test. Χ2 was used to analyze the frequency differences. The results were considered statistically significant if the *p*-value was less than 0.05.

## 3. Results

### 3.1. Quality of Life and Sleep Disturbances in Psoriasis

The QoL of almost all patients was impaired. The mean DLQI in the entire population was 12.7 ± 7.9 points. Based on DLQI scoring, the majority of patients demonstrated a very large effect (*n* = 108; 36.7%) or extremely large effect (*n* = 69; 26.5%) of psoriasis on QoL. Only 22 subjects (7.5%) reported no influence (DLQI scoring: 0 or 1) and 44 (15.0%) reported a small effect (DLQI: 2–5 points) of psoriasis on QoL. Comparison of DLQI scoring between different clinical psoriasis subtypes revealed the highest impact on QoL in the erythrodermic psoriasis variant (mean DLQI: 18.1 ± 6.9; *p* < 0.001 compared to other subtypes; Figure 1). There were no significant differences in the remaining psoriasis subgroups regarding QoL (Figure 1). The degree of the impact of psoriasis on QoL in different clinical variants according to DLQI score were summarized on Figure 2.

Regarding sleeping problems, 39.3% (*n* = 116) of patients reported occasional difficulties in falling asleep, and 22.7% of them (*n* = 67) had such problems almost every day. Moreover, 20.3% (*n* = 60) woke up during sleep almost every night, and a further 33.6% (*n* = 99) reported such problem sporadically. However, the majority of participants (79.0%) did not take any sleeping medications. We did not find any differences regarding sleeping problems and clinical variants of psoriasis (data not shown).

### 3.2. Factors Influencing Quality of Life

The statistical analysis between skin lesion severity indices (PASI, BSA, sPGA, PPSI, GPPSI) and the DLQI indicated that patients with more severe psoriasis had more impaired QoL (Table 2). Higher DLQI scoring indicating a more impaired QoL was also observed in patients suffering from pruritus within previous three days compared to subjects without pruritus (DLQI in the group with pruritus: 13.5 ± 7.5 points vs. DLQI in the group without pruritus: 8.0 ± 8.0 points; *p* < 0.001), with the highest DLQI scoring observed in patients with generalized pruritus (mean DLQI: 16.9 ± 7.6 points). As shown in Table 2, DLQI correlated with all pruritus intensity measurements (NRSmax, NRSaverage, 10-PSS). Interestingly, a low, albeit significant, correlation was also observed between DLQI and BMI (Body Mass Index: ρ = 0.17, *p* = 0.004). However, DLQI scoring was independent of gender (females: 12.4 ± 7.7 points vs. males: 12.8 ± 8.1 points, *p* = 0.66), age (ρ = 0.03, *p* = 0.61), psoriasis duration (ρ = 0.03, *p* = 0.6) and the coexistence of psoriatic arthritis (with psoriatic arthritis: 13.4 ± 8.7 points vs. without psoriatic arthritis: 12.6 ± 7.8 points, *p* = 0.59).

The marked influence of pruritus intensity on QoL may be, at least partially, explained by sleeping problems caused by pruritus. Difficulties in falling asleep, occurring even rarely, similarly to awakenings during the sleep, meaningfully impacted the QoL (Table 3). Patients using sleeping pills also had more reduced QoL than those who did not (Table 3).

The multiple regression analysis considering all above-mentioned parameters indicated that the involvement of body area by psoriasis (BSA), intensity of pruritus measured with 10-PSS and the occurrence and frequency of awakenings during the night caused by itching are all independent factors determining QoL in psoriatic patients.

### 3.3. Relationship between Pruritus and Sleep Disturbance

The sleeping problems reported were linked with pruritus intensity. Patients who had more severe pruritus measured with 10-PSS, NRSmax, and NRSaverage more commonly reported difficulties in falling asleep, awakenings from sleep and the need to use sleeping pills (Table 4). All differences between compared groups were significant (*p* < 0.001 for all comparisons in-between the subgroups).

### 3.4. Correlation between Severity of Pruritus and Quality of Life and Sleep Disturbances in Different Clinical Subtypes of Psoriasis

The intensity of pruritus was correlated with impairment of the QoL and the QoL impairment in patients with psoriasis of the scalp and palmoplantar pustular psoriasis. Surprisingly, QoL of patients suffering from erythrodermic and small-plaque psoriasis did not depend on the severity of itching (Table 5).

The clinical subtype of psoriasis was not related to difficulties with falling asleep (*p* = 0.83), awakenings during the night (*p* = 0.84), or the need to use the sleeping pills (*p* = 0.34).

## 4. Discussion

Our cross-sectional, binational study, including a wide spectrum of clinical subtypes of psoriasis, demonstrated the influence of the disease severity on the QoL, irrespective of the particular psoriasis variant. The mean DLQI scoring in our cohort of patients was 12.7 points, indicating on average a large effect of psoriasis on patients’ QoL. In the past, many researchers also investigated the QoL of psoriasis patients and obtained similar results demonstrating that psoriasis is a chronic dermatosis which exerts a profound effect on patients’ well-being [21,22,23,24,25]. However, in other studies, physicians were focused on various factors that might play a crucial role in lowering the QoL such as the presence of pruritus, occupation of the anogenital area by the disease or visibility of skin lesions [24,25,26,27]. Data comparing QoL in different clinical subtypes of psoriasis are quite limited, as the previously published reports are predominated by the patients suffering from plaque-type psoriasis, which is the most common one. Here, we aimed to analyze whether the level of QoL impairment may differ between various psoriasis variants. We also compared included participants with various psoriasis subtypes regarding sleeping problems.

As presented above, patients with erythrodermic psoriasis had a lower QoL than subjects with other disease subtypes. Systemic symptoms such as chills and fever together with malaise and weakness may be a good explanation for the low QoL in erythrodermic patients, but other factors may also play a role. Pruritus is one of the most relevant clinical parameters that may determine the degree of QoL impairment. In concordance with previous studies [27,28,29,30,31,32], our data confirmed that pruritus in psoriasis is very frequent, reaching the prevalence of up to 100% in some psoriasis variants, e.g., small-plaque-type psoriasis, scalp psoriasis, and generalized pustular psoriasis [9]. The severity of pruritus correlated with the impairment of QoL, particularly in scalp psoriasis and palmoplantar pustular psoriasis. This observation is slightly surprising as these subtypes were characterized by the lowest mean DLQI scoring (9.8 and 10.8, respectively) among all analyzed subgroups.

QoL was decreased in patients with sleeping disorders. The importance of good sleep for overall well-being is well documented and an insufficient amount of high-quality sleep affects many aspects of life [4]. In our study, the most frequent complaint was difficulty with falling asleep, signaled by 70% of patients. Additionally, more than half of them had to wake up from sleep during the night. In the majority of participants, sleeping problems usually did not appear to be strong enough to result in it being necessary to use sleeping pills. As found in our study, itching can directly affect sleep and patients with more severe pruritus had more difficulties in falling asleep, more frequently report awakenings during the night and more commonly needed to use the sleeping pills. Jensen et al. obtained similar results, and identified the intensity of pruritus as the major predictor of sleep impairment [33]. Hawro et al. [5] also noted that the difficulty in falling asleep was related to itching and explained this by the coexistence of negative emotions such as irritation and discomfort accompanying the pruritus. However, in this study, pruritus intensity did not appear to be sufficiently strong to result in awakenings during the night, which was contrary to our results.

Our study had some limitations. At first, only the DLQI was used to assess QoL, without a QoL-specific instrument for psoriasis. However, DLQI is a widely used method of QoL assessment and in many studies it is the major way how QoL impairment is assessed. Furthermore, sleeping disorders were diagnosed on the basis of three simple questions instead of a validated questionnaire, such as Athens Insomia Scale or Pittsburgh Sleep Quality Index [34,35]; however, we have used them successfully in our previous studies in patients suffering from pruritus [9,36,37] and they were used also by other authors [38,39].

## 5. Conclusions

QoL and the sleep quality of patients suffering from different subtypes of psoriasis are markedly decreased. The healthcare systems should aim to holistically manage psoriatic patients. To make it possible there is a need to focus not only on the objective severity of the disease but also on its impact on the QoL and sleep, as well as on subjective symptoms. Identifying the most disturbed area of life and recognizing the most bothersome symptoms of psoriasis are pivotal to choose the right treatment and achieve a therapeutic success. Pruritus should always be considered as one of key factors determining QoL, and its reduction should always be an important therapeutic goal.

## Figures and Tables

**Figure 1 jcm-11-05553-f001:**
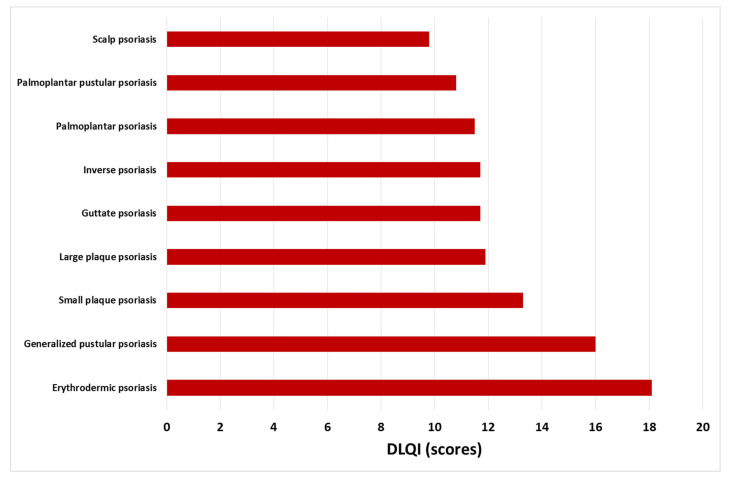
Mean Dermatology Life Quality Index score by clinical subtypes of psoriasis.

**Figure 2 jcm-11-05553-f002:**
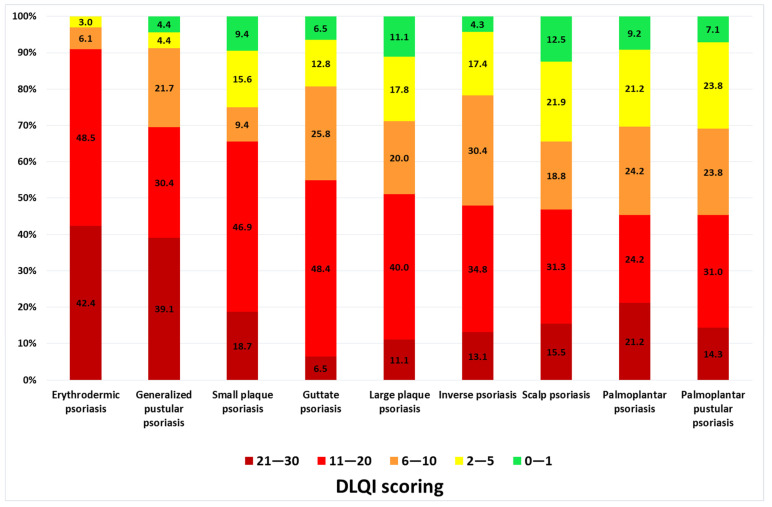
Degree of psoriasis impact on quality of life based on Dermatology Life Quality Index (DLQI) scoring in various clinical variants of psoriasis (0–1—no effect, 2–5—small effect, 6–10—moderate effect, 11–20—very large effect, 21–30—extremely large effect on patient’s life).

**Table 1 jcm-11-05553-t001:** Comparison of demographic, anthropometric and clinical data in patients with various clinical subtypes of psoriasis (statistically significant differences from other patients (*p* < 0.05) were marked with “*”).

	Number of Subjects (%)	Gender Female (%)/Male (%)	Age Min–Max (Mean ± SD)	BMI Min–Max (Mean ± SD)	Age at Disease Onset Min–Max (Mean ± SD)	Coexisting Psoriatic Arthritis (%)
**All subjects**	295 (100)	148 (50.2)/147 (49.8)	16–77 (45.0 ± 15.3)	14.5–46.0 (27.5 ± 5.6)	2–76 (32.9 ± 16.8)	27 (9.2)
**Large plaque psoriasis**	45 (15.3)	15 (33.3) 30 (66.7)	17–77 (46.7 ± 15.9)	21.4–42.8 (30.2 ± 5.5) *p* < 0.001 *	6–72 (30.3 ± 17.5)	5 (11.1)
**Small plaque psoriasis**	32 (10.8)	11 (34,4) 21 (65.6)	16–69 (40.6 ± 13.6)	14.5–37.6 (26.4 ± 5.3)	3–58 (27.3 ± 15.1)	2 (6.3)
**Guttate psoriasis**	31 (10.5)	14 (45.2) 17 (54.8)	17–73 (37.5 ± 11.6)	19.5–46.0 (27.5 ± 6.4)	7–53 (24.0 ± 12.1)	4 (12.9)
**Palmoplantar psoriasis**	33 (11.2)	18 (54.6) 15 (45.4)	16–71 (45.3 ± 14.9)	18.1–43.4 (28.3 ± 5.7)	10–69 (36.1 ± 16.5)	2 (6.1)
**Scalp psoriasis**	32 (10.8)	19 (59.4) 13 (40.6)	16–61 (34.9 ± 12.4)	15.4–34.1 (24.2 ± 4.4) *p* < 0.001 *	6–50 (23.7 ± 11.7)	2 (6.3)
**Inverse psoriasis**	23 (7.8)	12 (52.2) 11 (47.8)	18–67 (41.5 ± 14.0)	18.8–41.5 (29.3 ± 5.6)	13–60 (30.7 ± 14.2)	2 (8.7)
**Erythrodermic psoriasis**	33 (11.2)	9 (27.3) 24 (72.7)	18–71 (47.5 ± 16.6)	17.6–44.5 (27.8 ± 6.4)	3–60 (31.9 ± 16.7)	5 (15.2)
**Palmoplantar pustular psoriasis**	43 (14.6)	37 (86.1) 6 (13.9)	28–77 (53.8 ± 12.6) *p* < 0.001 *	18.8–37.2 (26.4 ± 4.5)	16–69 (45.7 ± 12.1) *p* < 0.001 *	1 (2.3)
**Generalized pustular psoriasis**	23 (7.8)	13 (56.5) 10 (43.5)	28–76 (54.9 ± 14.3) *p* < 0.001 *	17.2–39.3 (27.0 ± 4.8)	2–76 (45.5 ± 19.1) *p* < 0.001 *	4 (17.4)

**Table 2 jcm-11-05553-t002:** Correlations between Dermatology Life Quality Index (DLQI) and indices of psoriasis severity and pruritus intensity according to Spearman rank correlation test (10-PSS—10-item Pruritus Severity Scale, BSA—Body Surface Area, GPPSI—Generalized Pustular Psoriasis Severity Index, NRS_average_—average Numerical Rating Scale, NRS_max_—maximal Numerical Rating Scale, PASI—Psoriasis Area and Severity Index, PPSI—Palmoplantar Pustulosis Severity Index, sPGA—static Physician Global Assessment).

	DLQI		
	*n*	Spearman Rank Correlation Coefficient (ρ)	*p*
**PASI**	209	0.41	<0.001
**BSA**	281	0.38	<0.001
**sPGA**	293	0.37	<0.001
**GPPSI**	23	0.34	0.12
**PPSI**	42	0.52	<0.001
**NRS_max_**	274	0.39	<0.001
**NRS_average_**	274	0.34	<0.001
**10-PSS**	263	0.51	<0.001

**Table 3 jcm-11-05553-t003:** Comparison of quality of life in relation to the sleeping problems (DLQI—Dermatology Life Quality Index).

		*n*	Mean DLQI Scoring	*p*
Problems in falling asleep	Almost always	67	17.3 ± 7.1	<0.001
	Rarely	115	13.5 ± 7.0	
	Never	111	9.0 ± 7.5	
Awakenings during the night	Almost every night	60	17.4 ± 7.0	<0.001
	Rarely	98	13.9 ± 7.5	
	Never	135	9.7 ± 7.2	
Use of sleeping medication	Almost every day	10	18.4 ± 7.0	<0.001
	Rarely	51	15.5 ± 7.0	
	Never	232	11.8 ± 7.9	

**Table 4 jcm-11-05553-t004:** Comparison of pruritus intensity in relation to the sleeping problems (10-PSS—10-item Pruritus Severity Scale, NRS_average_—average Numerical Rating Scale, NRS_max_—maximal Numerical Rating Scale).

		Mean NRS_max_	*p*	Mean NRS_average_	*p*	Mean 10-PSS	*p*
Problems in falling asleep	Almost always	7.3 ± 2.2	<0.001	6.0 ± 2.5	<0.001	13.5 ± 3.5	<0.001
	Rarely	5.5 ± 2.2		4.1 ± 2.1		9.4 ± 3.1	
	Never	3.3 ± 2.7		2.6 ± 2.5		5.8 ± 3.8	
Awakenings during the night	Almost every night	7.3 ± 2.3	<0.001	6.1 ± 2.6	<0.001	13.3 ± 3.4	<0.001
	Rarely	5.6 ± 2.3		4.2 ± 2.2		9.7 ± 3.6	
	Never	3.8 ± 2.7		2.8 ± 2.4		6.5 ± 3.9	
Use of sleeping medication	Almost every day	8.2 ± 1.7	<0.001	7.4 ± 2.5	<0.001	15.1 ± 3.0	<0.001
	Rarely	6.4 ± 2.5		4.8 ± 2.3		10.7 ± 3.5	
	Never	4.7 ± 2.8		3.6 ± 2.6		8.3 ± 4.5	

**Table 5 jcm-11-05553-t005:** Correlations between pruritus intensity and quality of life according to the Spearman rank correlation test in different subtypes of psoriasis (10-PSS—10-item Pruritus Severity Scale, DLQI—Dermatology Life Quality Index, NRS_average_—average Numerical Rating Scale, NRS_max_—maximal Numerical Rating Scale).

		Large-Plaque Psoriasis	Small- Plaque Psoriasis	Guttate Psoriasis	Palmo-Plantar Psoriasis	Psoriasis of the Scalp	Inverse Psoriasis	Erythrodermic Psoriasis	Palmoplantar Pustular Psoriasis	Generalized Pustular Psoriasis
**DLQI vs.**	**NRS_average_**	**ρ = ** **0.51,** ***p* < 0.001**	ρ = 0.17, *p* = 0.37	ρ = 0.18, *p* = 0.35	ρ = 0.27, *p* = 0.12	**ρ =** **0.52,** ***p* = 0.003**	ρ = 0.17, *p* = 0.44	ρ = 0.15, *p* = 0.41	**ρ = ** **0.43,** ***p* = 0.004**	**ρ = ** **0.48,** ***p* = 0.02**
	**NRS_max_**	**ρ = ** **0.56,** ***p* < 0.001**	ρ = 0.07, *p* = 0.71	ρ = 0.3 *p* = 0.11	ρ = 0.29, *p* = 0.11	**ρ = ** **0.43,** ***p* = 0.01**	**ρ = ** **0.42,** ***p* < ** **0.05**	ρ = 0.25, *p* = 0.15	**ρ = 0.53,** ***p* < 0.001**	ρ = 0.34, *p* = 0.11
	**10-PSS**	ρ = 0.16, *p* = 0.29	ρ = 0.14, *p* = 0.43	**ρ = 0.54,** ***p* = 0.002**	**ρ = 0.51,** ***p* = 0.003**	**ρ = 0.64,** ***p* < 0.001**	ρ = 0.38, *p* = 0.08	ρ = 0.18, *p* = 0.32	**ρ = 0.56,** ***p* < 0.001**	**ρ = 0.41,** ***p* < 0.05**

## Data Availability

Data is contained within the article.

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
