# Peer review of "The Impact of Pruritus on the Quality of Life and Sleep Disturbances in Patients Suffering from Different Clinical Variants of Psoriasis"

_jcm, 2022, doi:10.3390/jcm11195553_

Round 1

Reviewer 1 Report

Interesting work with a good clinical message

1.     Shorten “have been reported to be” to “are”

2.     Delete “It was observed that” and similar useless verbiage.

3.     Get rid of the ambiguous words "significantly" and "significant." The reader might be misled into thinking that something that was not due to change was clinically meaningful. If you are using significant to refer to statistical significance, there's no need to be redundant; just show the p value. 

4.     Delete found from “was found in”

5.     None is singular.  Change “were no relevant” to “was no relevant”

6.     Shorten “turned out to be” to “was”

7.     The Abstract Results should be more quantitative.

8.     Delete “it is widely recognized that”.  Phrases ending in “that” are useless verbiage that should be deleted. 

9.     Don’t use a whole sentence to tell the reader there is a table. To refer to a table or figure, just put "(Table X)" or "(Figure x)" at the end of an appropriate sentence. 

10.  Delete “It was observed that”

11.  Shorten “Strong, statistically significant correlations were found between the intensity of pruritus and the impairment of the QoL” to “The intensity of pruritus was correlated with impairment of the QoL”

12.  Shorten “The detailed analysis of the correlations between severity of pruritus and QoL for each clinical psoriasis subtype was presented in Table 5” to “(Table 5).”

13.  Shorten “We did not observe any relationship between clinical subtype of psoriasis and…” to “clinical subtype of psoriasis was not related to…”

14.  Delete the entirely redundant sentence, “In other words, various clinical subtypes of psoriasis did not differ from the other subtypes regarding sleeping problems.”

15.  Make similar changes throughout the Discussion.

Author Response

We are grateful to the reviewer for the positive reception of our study and for all comments which helped us to improve our manuscript. We have revised the manuscript strictly according to your recommendation. All modifications have been highlighted in the corrected manuscript. 

Reviewer 2 Report

In this original study, the authors analyze the impact of itching on quality of life and sleep quality in patients with different varieties of psoriasis.

Although it is a relatively new topic of study, in my opinion there are important methodological limitations that invalidate the study. The main one is the use of a method NOT validated to measure the quality of sleep. Since it is one of the main objectives of the study, it cannot be replaced or improved once the study has been carried out. The authors indicate this is the limitations but it is not enough just to indicate it.

On the other hand, the inclusion/exclusion criteria are poorly defined, with only a few comments on the excluded patients.

What is nummular psoriasis?

How have you classified the different forms of psoriasis? Very frequently, patients with psoriasis vulgaris have scalp involvement, for example. Given that another of the objectives is to differentiate the affectation of pruritus according to different subtypes of psoriasis, this is another essential aspect.

Author Response

We are grateful to the reviewer for valuable comments. Below we have provided answers to the criticism: 

Reviewer: Although it is a relatively new topic of study, in my opinion, there are significant methodological limitations that invalidate the study. The main one is using a method NOT validated to measure the quality of sleep. Since it is one of the main objectives of the study, it cannot be replaced or improved once the study has been carried out. The authors indicate this is the limitation but it is not enough just to indicate it.

Authors: Indeed, we have not validated our questions regarding sleep problems, which was indicated as a limitation of the study. However, the same three questions were used not only by us in previously conducted studies but were also used by other authors, e.g.  Huet F, Faffa MS, Poizeau F, Merhand S, Misery L, Brenaut E. Characteristics of Pruritus in Relation to Self-assessed Severity of Atopic Dermatitis. Acta Derm Venereol. 2019 Mar 1;99(3):279-283.,  Sánchez-Pérez J, Daudén-Tello E, Mora AM, Lara Surinyac N. Impact of atopic dermatitis on health-related quality of life in Spanish children and adults: the PSEDA study. Actas Dermosifiliogr. 2013 Jan;104(1):44-52. English, Spanish. doi: 10.1016/j.ad.2012.03.008. Epub 2012 Jul 28. PMID: 22841507. (added to the revised manuscript) 

Therefore, we have the feeling, that although not formally validated by our group, these 3 questions can be considered a solid measurement of sleep problems. 

Reviewer: On the other hand, the inclusion/exclusion criteria are poorly defined, with only a few comments on the excluded patients.

Authors: We have provided more detailed inclusion/exclusion criteria. 

Reviewer: What is nummular psoriasis?

Authors: In our country, we have used this term for small plaque psoriasis. For clarity, we have changed the name from nummular into small plaque psoriasis according to Griffiths et al. A classification of psoriasis vulgaris according to phenotype. Br J Dermatol. 2007 Feb;156(2):258-62.  

Reviewer: How have you classified the different forms of psoriasis? Very frequently, patients with psoriasis vulgaris have scalp involvement, for example. Given that another of the objectives is to differentiate the affectation of pruritus according to different subtypes of psoriasis, this is another essential aspect.

Authors: Classification of psoriasis was done according to Griffiths et al. A classification of psoriasis vulgaris according to phenotype. Br J Dermatol. 2007 Feb;156(2):258-62.  We have selected only these patients which could be allocated to one of the predefined psoriasis subtypes. Other patients were excluded as our aim was to compare pruritus across various subtypes of psoriasis. 

Round 2

Reviewer 2 Report

Thank you very much for the review. The authors have responded point by point to the comments. In my opinion, the work would have been better if a validated questionnaire had been used. But it is true that there is experience in the questions they have used, it has been cited in the text and it appears in the limitations section. The rest of the considerations about the classification have also been answered.